# Use of Phase-Angle Model for Full-Field 3D Reconstruction under Efficient Local Calibration

**DOI:** 10.3390/s24082581

**Published:** 2024-04-18

**Authors:** Fengxiao Lei, Ruijie Ma, Xinghui Li

**Affiliations:** 1Tsinghua Shenzhen International Graduate School, Tsinghua University, Shenzhen 518055, China; leifx21@mails.tsinghua.edu.cn (F.L.); mrj22@mails.tsinghua.edu.cn (R.M.); 2Tsinghua-Berkeley Shenzhen Institute, Tsinghua University, Shenzhen 518055, China

**Keywords:** calibration, fringes projection, 3D reconstruction

## Abstract

Currently, 3D reconstruction methods in structured light are generally implemented in a pre-calibrated area. To realize a full-field reconstruction, the calibration plate can be moved to multiple positions in a time-consuming manner, or the whole field can be calibrated with the help of a large calibration plate, which is more costly. In this paper, we address this problem by proposing a method for obtaining a global phase-angle model under a locally calibrated region, and based on this relationship, we investigate and analyze the reconstruction inside and outside of the calibrated zone. The results show that the method can reconstruct the object outside of the calibration zone completely, and can keep the planarity error around 0.1 mm and the sphericity error below 0.06 mm. The method only requires local calibration of the projected fringes at the two calibration positions to realize the 3D reconstruction of the full-field, which makes the method more advantageous.

## 1. Introduction

Fringe projection profilometry (FPP) has emerged as a powerful and versatile technique in the field of optical metrology and surface profilometry. Over the years, it has gained significant attention due to its high-accuracy [1,2], high-speed [3,4], and high-resolution [5,6] capabilities, enabling the accurate measurement of 3D shapes and surface topography. This technology has been widely applied in industrial measurement [7], cultural relic protection [8], surgical medicine [9], artificial intelligence [10], and other fields [11,12]. Its implementation process mainly consists of the following steps: the projector projects regular sinusoidal fringe patterns; then, the camera captures the modulated fringe patterns, and after phase calculation, 3D reconstruction is achieved with the help of system calibration parameters [13].

Since the calibration of a FPP system determines in which form the 3D reconstruction is carried out, it is closely related to the accuracy of the reconstruction and can affect the range of measurement to a certain extent. Different calibration strategies [14,15,16,17,18] have been widely and deeply investigated. Currently, there are mainly two classical calibration technology routes, including the stereo-vision-based calibration method and the phase-coordinate mapping model-based calibration method [16]. In the stereo-vision-based FPP system, the projector is regarded as an inverse camera [19,20], so it can search for matching points in a binocular framework. However, this type of calibration method is not applicable to the current emerging structured light projection devices, such as MEMS (micro-electro-mechanical system)-mirror-based projectors, that can only project unidirectional sinusoidal fringes because they need to determine the pixel positions of the markers of the calibration plate in the DMD image with the help of the orthogonal absolute phase. In addition, this type of method requires more computational resources for homologous point matching, which limits the extended use of this method. In the phase-coordinate mapping model [14,15,21,22], the coordinates X, Y, and Z can be directly obtained by bringing the absolute phase into a pre-calibrated function, which does not require orthogonal phases and thus is still applicable in FPP systems composed of MEMS-mirror-based projectors.

However, in the phase-coordinate mapping model phase, the direct independent variable generated by 3D coordinates needs to be pre-calibrated with points at known positions in space. Initially, the coordinates of these known points were represented in the world coordinate system, which meant that precise translation stages were needed to control the transformation of their coordinates, and these points were clearly always on the calibration board. Some existing methods [14,15] have relaxed restrictions and do not require precision translation stages or gage blocks for calibration. The coordinates of these points appear in the camera coordinate system, and they need to provide raw calibration data for the fitting of the phase-coordinate mapping model together with the phase of the corresponding pixels on the calibration board plane. After analysis, it can be found that both the phase and coordinates need to be located on the calibration board to participate in the calibration of the mapping model. This means that a reliable mapping model cannot be formed outside of the coverage range of the calibration board, which leads to the failure of 3D reconstruction.

Recently, researchers have proposed new characterization models to reveal the physical processes of unidirectional fringes projection, including plane models and ray models, etc. Yang et al. [17] have proposed a seven-parameter curved light surface model for unidirectional fringe projection to characterize non-ideal projections. However, this method requires the checkerboard to be large enough to allow all 1024 curved light surfaces to be projected onto it to obtain the 3D coordinates of enough spatial points, which means that a small calibration plate cannot efficiently create a global light surface model. Miao et al. [14] have proposed a new isophase plane model, which evolves into a reciprocal polynomial mapping model by merging calibration parameters. A look-up table (LUT) approach is used to achieve fast and highly accurate measurements; however, the coverage of the LUT is still limited to the calibration plate. Similar LUT mapping methods used in reference [13] have been tried, which implies that they have the same limitations. Wan et al. [18] have designed a large-scale calibration strategy based on the method of linear interpolation in the light plane. But, the spatial equation of the remaining light planes can only be calculated when it is between the calibrated light planes. To ensure the reliability of large-scale reconstruction, 11 sets of calibration data are needed to ensure full coverage of the projection range, which is inefficient. In addition to the plane model, the projection process can also be refined into a ray model. Yang et al. [16] have achieved 3D reconstruction using triangulation constructed from both the projected rays and the camera. However, each ray also needs to be constructed in the LUT, which means that the equation for the ray can only be pre-estimated within the local area covered by a calibration board. At present, the calibration strategies in unidirectional fringe projection profilometry mainly focus on the reconstruction effect within the calibration area. Inferring 3D reconstruction information beyond the range of the calibration plate in reconstructing full-field scenes has not yet received in-depth research and focused attention.

The phase-angle model [23] is a novel representation model applied in unidirectional fringe projection profilometry, and it can be used to describe the angle variation relationship between phase and isophase planes in space, providing a new approach for 3D reconstruction. The isophase plane represents a hypothetical plane formed by coordinates in projection space that are different in position but still have the same phase. In structured light systems based on MEMS-mirror-based projectors, the isophase plane can be understood as a plane formed by the reflection of laser vertical lines through the MEMS mirror. In the system composed of DLP projectors, the isophase plane is located in the direction reflected by a column of small micro mirrors in the array. After obtaining the phase-angle model, the position of the current phase plane can be located through phase positioning, and then the 3D coordinates can be obtained by combining the geometric constraints of the camera perspective instead of using phase as the direct independent variable to participate in coordinates generation. This approach avoids the problem of 3D coordinate-generation errors when the phase-coordinate mapping model is not fully fitted.

In this paper, a reconstruction method using a phase-angle model to infer 3D information outside of the calibration zone is proposed. It does not require the calibration plate to be large, but requires that the calibration plate positions of two projection fringes have overlapping areas to construct the relationship between the phase and the deflection angle of the isophase plane, which is then used to locate the isophase plane within the projection space rather than only within the calibration plate area. This endows the 3D reconstruction capability beyond the calibration zone. We have validated the feasibility of the method through experiments, comparing the reconstruction inside and outside of the calibration zone using similarity in reconstructed morphology and reconstruction accuracy. We have also compared the proposed method with existing phase-coordinate mapping model methods, demonstrating its advantages in reconstructing 3D information outside of the calibration zone. In addition, by reducing the size of the calibration area and comparing the reconstruction accuracy vertically, the results show that this method still has robust 3D reconstruction ability.

## 2. Principle

### 2.1. Camera Imaging Model

The camera’s imaging mechanism operates through perspective projection. During forward projection, the projection center casts points from the object onto the imaging plane, producing the camera’s image. In contrast, 3D reconstruction involves the inverse process of back-projecting image points to pinpoint their corresponding spatial coordinates. This process can be expressed mathematically as follows:(1)$$s_{c}\left[\begin{array}{c} u\\ v\\ 1 \end{array}\right]=K_{c}\left[\begin{array}{c} x^{c}\\ y^{c}\\ z^{c}\\ 1 \end{array}\right]=K_{c}\left[\begin{array}{cc} R & T\\ 0 & 1 \end{array}\right]\left[\begin{array}{c} x^{w}\\ y^{w}\\ z^{w}\\ 1 \end{array}\right],\;K_{c}=\left[\begin{array}{c} \begin{array}{cccc} f_{u} & \gamma & u_{0} & 0\\ 0 & f_{v} & v_{0} & 0\\ 0 & 0 & 1 & 0 \end{array} \end{array}\right].$$
where the coordinates of point ***P*** in the world coordinate system (WCS) and camera coordinate system (CCS) are (*x^w^*, *y^w^*, *z^w^*) and (*x^c^*, *y^c^*, *z^c^*), respectively; rotation matrix ***R*** and a translation vector ***T*** represent the rigid body transformation between the WCS and the CCS; *s_c_* stands for the scaling factor; ***K****_c_* are the camera’s internal parameters, where *f_u_* and *f_v_*, respectively, denote the equivalent focal lengths in the horizontal and vertical directions; *γ* denotes the skew factor between the *u* axis and *v* axis in the imaging plane; and (*u*_0_, *v*_0_) represents the principal point of the pixel plane. When we use (*k*_1_, *k*_2_, *k*_3_) to denote the radial aberration coefficients, (*p*_1_, *p*_2_) to denote the tangential aberration coefficients, $\left({\tilde{u}}_{d},{\tilde{v}}_{d}\right)$ and $\left(\tilde{u},\tilde{v}\right)$ to denote the normalization terms for the actual imaging point $\left(u_{d},v_{d}\right)$ and the ideal imaging point (*u*, *v*), respectively, we can take the lens aberration into account with the following equations:(2)$$\left\{\begin{array}{c} {\tilde{u}}_{d}=\tilde{u}(1+k_{1}r^{2}+k_{2}r^{4}+k_{3}r^{6})+[2p_{1}\tilde{u}\tilde{v}+p_{2}(r^{2}+2{\tilde{u}}^{2})]\\ {\tilde{v}}_{d}=\tilde{v}(1+k_{1}r^{2}+k_{2}r^{4}+k_{3}r^{6})+[2p_{2}\tilde{u}\tilde{v}+p_{1}(r^{2}+2{\tilde{v}}^{2})]\\ r^{2}={\tilde{u}}^{2}+{\tilde{v}}^{2} \end{array}\right..$$

### 2.2. Basis of the Phase-Angle Model

In Figure 1, if the calibration board cannot fully cover the projection range in practical application scenarios for the reconstruction method of phase-coordinate mapping model, only the local calibration zone can participate in the reconstruction, which limits its reconstruction range ability for systems with large projection or field of view ranges. The phase-angle model mentioned below can cover all points in the projection field, which means that in the space outside of the calibration area, the phase-angle model can infer and assist in solving reconstruction problems cleverly.

It is worth noting that the definition of the deflection angle of the isophase plane requires a reference position, which can be taken arbitrarily but is fixed after the system calibration is completed. As shown in Figure 1, the green dashed line represents the position of the reference isophase plane, corresponding to phase *φ_ref_*, and its position can be arbitrarily taken within the projection range. Assuming the working distance of the projector is *d*, let *O_x_* be the origin of the *x* axis. The reference phase *φ_ref_* corresponds to *x*_1_ on the *x* axis, and the angle between the reference isophase plane and the straight line *AO_x_* is *θ*_1_. The current phase *φ_cur_* corresponds to *x_cur_*, and the angle between its isophase plane and the reference isophase plane is *θ′*. At the same time, its angle with the straight *AO_x_* is *θ*_2_.

For sinusoidal fringes projected on the orthogonal virtual plane, the linear relationship *φ = kx + b* is satisfied between the spatial coordinates *x* and the unwrapped phase *φ* because the spacing of the fringes is uniformly distributed. Then, it is clear that the reference phase *φ_ref_* and the spatial coordinates *x*_1_, as well as the current phase *φ_cur_* and the corresponding spatial coordinates *x_cur_,* satisfy this linear relationship:(3)$$\left\{\begin{array}{c} x_{1}=\frac{\varphi_{ref}-b}{k}\\ x_{cur}=\frac{\varphi_{cur}-b}{k} \end{array}\right..$$

Based on all of the parameters available, the following relationship between angles and coordinates can be found:(4)$$\left\{\begin{array}{c} \tan\theta \prime =\tan(\theta_{1}+\theta_{2})=\frac{\tan\theta_{1}+\tan\theta_{2}}{1-\tan\theta_{1}\cdot \tan\theta_{2}}\\ \tan\theta_{1}=-\frac{x_{1}}{d}\\ \tan\theta_{2}=\frac{x_{cur}}{d} \end{array}.\right.$$

Associating Equations (3) and (4), eliminating *θ*_1_, *θ*_2_, *x*_1_, and *x_cur_*, and combining the correlation coefficients yields the equation for the relationship between the deflection angle *θ*′ and the current phase *φ_cur_*, which is the algebraic expression for the phase-angle model:(5)$$f_{\varphi }=\tan\theta \prime =\frac{\varphi_{cur}-\varphi_{ref}}{kd+\frac{\left(\varphi_{ref}-b)\cdot (\varphi_{cur}-b\right)}{kd}}=\frac{\varphi_{cur}-\varphi_{ref}}{a_{1}\cdot \varphi_{cur}+a_{2}}.$$
where *φ_ref_* is a known quantity, a pre-determined value of a reference phase, and *a*_1_ and *a*_2_ are the model parameters to be calibrated.

In the process of proposing the phase-angle model, there must theoretically exist a plane perpendicular to the projection direction of the projector, on which the sinusoidal fringes are uniformly distributed. In practical application of the phase-angle model, the position of the calibration plane is not usually guaranteed to be orthogonal, especially when manually changing the position of the calibration plate. At this time, the projected fringes on the inclined calibration plane will appear as changes in the degree of sparsity. As shown in Figure 2a, with respect to the geometrical position of the inclined calibration plane and the orthogonal calibration plane in the system, the projected fringes on the side close to the projector are more tightly packed and have higher spatial frequency, while the fringes on the side far from the projector are sparser and the spatial frequency becomes lower. Then, as the change in spatial frequency occurs it results in the phase on the plane no longer showing a linear change trend. Therefore, it is necessary to prove that the corresponding phases of the same light plane projected on different positions of the inclined calibration plate are still equal, which guarantees the feasibility of the phase-angle model.

In the top view of the position of the calibration plane in Figure 2b, we can assume that there is an angular skew $\alpha_{s1}$ between an inclined calibration plane and the orthogonal position, which intersect at a point $O_{1}$, and that the distance from the point $O_{1}$ to the projector is *L*. Within the projection range, there is a projection plane with an angle of θ from the normal projection direction, which intersects with the orthogonal calibration plane at *A*. In addition, the angle between this projection plane and the orthogonal calibration plane is β, as shown in Figure 2b. The outgoing light plane intersects with the inclined calibration plane at point *B*, with an angle of $∠O_{1}BA$ = γ. When points *A* and *B* are defined on the right side of $O_{1}$, it is specified that |*O*_1_*B*| = *x_skew_* and |*O*_1_*A*| = *x_p_*, so there exists the following geometric relationship:(6)$$\left\{\begin{array}{c} \beta =90^{\circ }\;-\;\arctan(\frac{x_{p}}{L})\\ \gamma {=90}^{\circ }\;-\;\arctan(\frac{x_{p}}{L})\;-\;\alpha_{s1}\\ \frac{x_{p}}{\sin\gamma }=\frac{x_{skew}}{\sin(180^{\circ }-\beta )} \end{array}\right..$$

By combining Equation (6), the equation relationship of *x_skew_* with respect to *x_p_* can be obtained:(7)$$x_{skew}=\frac{x_{p}}{\cos\alpha_{s1}-\frac{x_{p}}{L}\sin\alpha_{s1}}.$$
when *A* and *B* are on the left side of $O_{1}$, it is specified that |*O*_1_*B*| = −*x_skew_* and |*O*_1_*A*| = −*x_p_*, and the same derivation method can be used to obtain the following:(8)$$-x_{skew}=\frac{-x_{p}}{\cos\alpha_{s1}+\frac{-x_{p}}{L}\sin\alpha_{s1}},$$

Equation (8) is expressed in the same form as Equation (7), so that the relationship between the transverse coordinate *x_p_* on the orthogonal calibration plane and the transverse coordinate *x_skew_* on the inclined calibration plane are related only to the angular skew $\alpha_{s1}$ and the working distance *L*.

Assuming there is a (*x_p_*, *y*) point on the orthogonal calibration plane, the phase principal value can be obtained by solving the phase using the phase shift method at that point:(9)$$\phi \;=\;\arctan\left(\frac{-\sum_{i=1}^{N}B\cos(2\pi f_{0}x_{p}+\delta_{i})\sin[2\pi (i-1)/N]}{\sum_{i=1}^{N}B\cos(2\pi f_{0}x_{p}+\delta_{i})\cos[2\pi (i-1)/N]}\right).$$

Among them, *N* represents the number of phase shift steps, *B* represents the modulation intensity, *f*_0_ is the spatial frequency of sinusoidal fringes, and *δ_i_* is the phase shift increment of the *i*-th phase shift graph, which can be expressed as *δ_i_* = 2π (*i* − 1)/*N* + *φ*_0_, where *φ*_0_ represents the initial phase that can be artificially specified.

On the inclined calibration plane, according to the mapping relationship of the coordinates, the fringes will be distorted and the spatial frequency will show corresponding changes: *f* = [cos(*α_s_*_1_) − (*x_p_*/*L*) sin(*α_s_*_1_)] *f*_0_. At this point, the variation in light intensity with *x_skew_* on the inclined calibration plane can be obtained:(10)$$I_{i}^{\prime }(x_{skew},y)=c(x_{p},y)[A+B\cos(2\pi fx_{skew}+\delta_{i})],\;i=1,2,3,\ldots ,N.$$
where *c*(*x_p_*, *y*) represents the attenuation factor of the grayscale value that occurs when the position of point (*x_p_*, *y*) corresponds to the inclined calibration plane. According to the calculation method using the phase principal values, the phase principal values on the inclined calibration plane can also be obtained, and *f* and *x_skew_* can be substituted to obtain Equation (11):(11)$$\begin{array}{ll} \phi \prime \; & =\;\arctan\left(\frac{-\sum_{i=1}^{N}c(x_{p},y)\;[A+B\cos(2\pi fx_{skew}+\delta_{i})]\sin[2\pi (i-1)/N]}{\sum_{i=1}^{N}c(x_{p},y)\;\;[A+B\cos(2\pi fx_{skew}+\delta_{i})]\cos[2\pi (i-1)/N]}\right)\\ & =\;\arctan\left(\frac{-\sum_{i=1}^{N}\;B\cos(2\pi f_{0}x_{p}+\delta_{i})\sin[2\pi (i-1)/N]}{\sum_{i=1}^{N}\;B\cos(2\pi f_{0}x_{p}+\delta_{i})\cos[2\pi (i-1)/N]}\right). \end{array}$$

It can be seen from above that the phase principal value *ϕ*′ on *x_p_* is equal to the corresponding phase principal value *ϕ*′ on *x_skew_*. The corresponding absolute phases *φ* and *φ*′ are also equal, and the two cause phase mismatch due to factors such as projection distance, tilt angle, and intensity attenuation factor. It can be concluded that when the fringe frequency changes due to the tilt of the calibration plane in the projection space, the corresponding phases of the same light plane projected at different positions remain the same, and the phase-angle model is still feasible.

## 3. System Calibration and 3D Reconstruction

### 3.1. Local Calibration

In order to obtain the phase-angle model relationship, the calibration process first needs to obtain the phase samples on the calibration plate. As shown in the system calibration diagram in Figure 3, the calibration plate may only occupy a small part of the projection range, and we need to sample enough absolute phase samples {*φ*_0_, *φ*_1_, *φ*_2_, …} in the limited range.

After that, to find the deflection angle corresponding to these absolute phase samples, it is necessary to locate their corresponding isophase planes. This process requires a total of three operations. Firstly, the pixels equal to the absolute phase samples are found on the two calibration plates, *A_P_* and *B_P_,* used for calibration. Since the calibration plate cannot completely occupy the camera image, it is necessary to set a mask to find the pixels with the same phase within the specified range. Secondly, according to the homography change relationship between the pixel plane and the calibration plate plane with *H_A_* and *H_B_*, and the rotation translation *RT* change relationship between the world coordinate system and the camera coordinate system, the coordinates of the pixel points on the plate are converted into 3D coordinates under the camera coordinate system. This operation can be expressed as the equation:(12)$$\left\{\begin{array}{c} \left[\begin{array}{c} a^{p}\\ b^{p}\\ 1 \end{array}\right]=H\left[\begin{array}{c} u^{p}\\ v^{p}\\ 1 \end{array}\right]\\ \left[\begin{array}{c} X^{p}\\ Y^{p}\\ Z^{p} \end{array}\right]=\left[\begin{array}{cc} R & T \end{array}\right]\left[\begin{array}{c} a^{p}\\ b^{p}\\ 0\\ 1 \end{array}\right] \end{array}\right..$$

Among them, (*u^p^*, *v^p^*) is the pixel retrieved from the phase map, the phase of which is equal to the given sample phase, and (*X^p^*, *Y^p^*, *Z^p^*) is the 3D coordinate of the corresponding point on the calibration board. Finally, based on the 3D coordinates of the identified sample points, the solution of the equation for the sampled isophase planes can be completed. This can be achieved by solving a homogeneous linear system of Equation (13):(13)$$\left[\begin{array}{cccc} \vdots & \vdots & \vdots & \vdots \\ X_{i}^{p} & Y_{i}^{p} & Z_{i}^{p} & 1\\ \vdots & \vdots & \vdots & \vdots \\ X_{M}^{p} & Y_{M}^{p} & Z_{M}^{p} & 1 \end{array}\right]\left[\begin{array}{c} A^{F_{0}}\\ B^{F_{0}}\\ C^{F_{0}}\\ D^{F_{0}} \end{array}\right]=0.$$
where $A^{F_{0}}$, $B^{F_{0}}$, $C^{F_{0}}$, and $D^{F_{0}}$ represent the equation coefficients of a certain sample’s isophase plane, while the isophase plane space equations of all other absolute phase samples can be solved according to these three operations.

Up to this point, the samples {*φ*_0_, *φ*_1_, *φ*_2_, …} of the absolute phases and the plane equations {***F***_0_, ***F***_1_, ***F***_2_, …} of their corresponding isophase planes have been obtained and they are all *N_φ_* in number. Here, we may choose the isophase plane ***F***_0_ as the reference and calculate the *N_φ_* − 1 of angles sample {$\theta_{\varphi 1}$, $\theta_{\varphi 2}$, …} based on the normal vector between the remaining isophase planes and the reference isophase plane ***F***_0_. At this point, the phase sample {*φ*_1_, *φ*_2_, …} and deflection angle samples {$\theta_{\varphi 1}$, $\theta_{\varphi 2}$, …} are used to solve the model using the least square method, and the calibration of parameters *a*_1_ and *a*_2_ in the phase-angle model is completed.

In addition, we found that the constructed isophase planes are all deflected around a virtual rotation centerline ***L***_0_, and that the extrapolation ability of the phase-angle model is reflected in the fact that the actual deflection angle can exceed the calibrated range when the isophase planes are based on this rotation centerline. Therefore, it is necessary to determine the position of the rotation centerline. Here, a vector ***d*** = (*v*_1_, *v*_2_, *v*_3_) and a point ***p*** = (*p*_1_, *p*_2_, *p*_3_) are used to uniquely determine the rotation centerline, where vector ***d*** represents the direction of the line and point ***p*** represents any point on the line. Since the rotation centerline is perpendicular to the normal vectors of the sampling isophase planes {***F***_0_, ***F***_1_, ***F***_2_, …} and is located on the isophase planes, the following relationship is satisfied:(14)$$\left\{\begin{array}{c} \left[\begin{array}{ccc} A^{F_{i}} & B^{F_{i}} & C^{F_{i}} \end{array}\right]\left[\begin{array}{c} v_{1}\\ v_{2}\\ v_{3} \end{array}\right]=[0]\\ A^{F_{i}}p_{1}+B^{F_{i}}p_{2}+C^{F_{i}}p_{3}+D^{F_{i}}=0 \end{array}\right..$$

Among them, $A^{F_{i}}$, $B^{F_{i}}$, $C^{F_{i}}$, and $D^{F_{i}}$ represent the equation coefficients of any sampling isophase planes. By solving the equation, the spatial position of the rotation centerline ***L***_0_ can be obtained.

In general, the calibration process can be divided into the following steps: step 1 is followed to obtain samples of absolute phase {*φ*_0_, *φ*_1_, *φ*_2_, …} and their corresponding isophase planes {***F***_0_, ***F***_1_, ***F***_2_, …}; step 2 is pe4rformed to complete the calibration of parameters *a*_1_ and a_2_ in the phase-angle model; step 3 is completed to obtain the spatial position of the rotation centerline ***L***_0_.

### 3.2. Reconstruction for Global Scope

During the reconstruction process, the phase-angle model and rotation centerline determined during the calibration process are utilized, and these calibration results can be used for 3D reconstruction of all global pixels. The specific flowchart is shown in Figure 4. Firstly, extract the phase of each pixel on the reconstructed scene’s phase map one by one, and substitute it into the phase-angle model to obtain the current deflection angle *θ*′. Based on the rotation centerline, the position of the reference isophase plane and angle *θ*′, determine the spatial equation ***F****_cur_*: $A_{cur}X+B_{cur}Y+C_{cur}Z+D_{cur}=0$ for the current isophase plane. In this process, several conditions need to be realized: the rotation centerline is located on the current isophase plane, the angle between the current isophase plane and the reference isophase plane is *θ*′, and the normal vector modulus composed of the first three coefficients of ***F****_cur_* can be artificially specified as 1. Specifically, the following equation system can be obtained:(15)$$\left\{\begin{array}{c} A_{cur}v_{1}+B_{cur}v_{2}+C_{cur}v_{3}=0\\ A_{cur}p_{1}+B_{cur}p_{2}+C_{cur}p_{3}+D_{cur}=0\\ A_{cur}A+B_{cur}B+C_{cur}C=\cos\theta \prime \\ {\left(A_{cur}\right)}^{2}+{\left(B_{cur}\right)}^{2}+{\left(C_{cur}\right)}^{2}=1 \end{array}\right..$$

Solve the equation system (15) to complete Part 1 of the reconstruction flowchart. In addition, the back-projection ray ***R*** corresponding to the current pixel will intersect with the normalization plane at a point $x_{n}={\left(\tilde{u},\tilde{v},1\right)}^{T}$, and any point on ***R*** can be represented as follows:(16)$$\left[\begin{array}{c} X\\ Y \end{array}\right]=Z\left[\begin{array}{c} \tilde{u}\\ \tilde{v} \end{array}\right].$$

At this point, Part 2 of the reconstruction has been completed. Finally, by combining the ***F****_cur_* equation and the ***R*** ray equation, the intersection coordinates of the two can be solved, that is, the coordinates of the reconstructed points:(17)$$\left\{\begin{array}{c} X=\frac{-\tilde{u}D_{cur}}{\tilde{u}A_{cur}+\tilde{v}B_{cur}+C_{cur}}\\ Y=\frac{-\tilde{v}D_{cur}}{\tilde{u}A_{cur}+\tilde{v}B_{cur}+C_{cur}}\\ Z=\frac{-D_{cur}}{\tilde{u}A_{cur}+\tilde{v}B_{cur}+C_{cur}} \end{array}\right..$$

## 4. Experiment and Analysis

### 4.1. Calibration Results

To verify the feasibility of the proposed method for reconstruction in non-calibrated areas, we conducted experiments on an FPP system based on a DLP projector. The projector in the system was the Texas Instrument pro6500-rgb-235, with a resolution of 1920 × 1080 and a refresh rate of 100 fps @ 8 bit. And the camera used was the Balser aca 2040-120 um grayscale camera, with a resolution of 2064 × 1544 and a frame rate of 100 fps. The working distance of the system was approximately 450 mm, and the field of view of the camera was approximately 200 mm × 150 mm. In addition, we used a calibration board of the symmetrical circle grid type, which included eight rows and eleven columns of 12 mm spaced marker points.

As shown in Figure 5a,b, to free up more space outside of the calibration board for testing the 3D reconstruction effect, we placed the calibration board in the lower left corner area. By setting a mask, phase samples are selected within the specified rectangular area, as shown in the phase maps in Figure 5c,d. The same-colored circles represent matching sample points located on the same isophase plane, which are used for calculating the space equation of the isophase plane. There are a total of 11 colors corresponding to 11 samples of isophase planes, as shown in Figure 5e, and these are used as samples for fitting the phase-angle model to infer the localization of reconstruction points in areas outside of the coverage range of the calibration board.

Following the previous calibration process, the reference phase and the equation for the reference isophase plane were determined based on the selected sample points. Then, by solving the deflection angles corresponding to the isophase planes one by one, the optimization of the phase-angle model was completed, and the parameters *a*_1_ and *a*_2_ were obtained. Finally, the rotation centerline was obtained using the least squares method, and the relevant data are listed in Table 1.

### 4.2. Reconstruction Results

After completing the calibration using this method, the effectiveness of 3D reconstruction was tested. Firstly, the correctness of the method was verified based on the structured light system of the DLP projector. Secondly, the same calibration strategy used to test the applicability on emerging structured light projection devices, such as the FPP system of MEMS-mirror-based projectors, was used. As shown in Figure 6, (a) represents the fringe pattern of the object in the DLP-based FPP system, and (b) represents the 3D point cloud of the object reconstructed using this method; (c) represent the fringe pattern of the object in the MEMS-based FPP system, and (d) represents the corresponding 3D point cloud of the object. From the completeness of the results and the similarity of the shapes, it can be seen that this calibration method can achieve 3D reconstruction on different FPP systems.

By utilizing the scanning characteristics of the phase-angle model, as long as the projected fringes can cover the object, 3D reconstruction can be achieved efficiently and accurately, regardless of where the object is in the camera’s field of view. To verify the speculated reconstruction ability of this method, we compared the reconstruction results of the same object placed in the covered and uncovered areas of the calibration board and presented the feasibility of the proposed method in an intuitive form. As shown in Figure 7, (a) and (b) represent the positions of the two calibration boards, respectively, and (c) is the overlapping area of the two calibration boards, also known here as the calibration zone. The subgraphs (d), (h), (l), and (p) are the tested objects inside the calibration zone, and the same objects are also placed outside of the calibration zone, as shown in subgraphs (f), (j), (n), and (r). The subgraphs (e), (g), (i), (k), (m), (o), (q), and (s) show their reconstructed point cloud results, and, visually, there is no significant difference in the point cloud of the same object.

To highlight the speculated reconstruction ability of the proposed method outside of the calibration zone, the phase-coordinate mapping model was also compared with the proposed method. In this experiment, we placed the measured object at the junction of the calibration area and the non-calibration area, projected the fringes, calculated the phase, and used these two techniques to reconstruct the object separately. The specific results are shown in Figure 8. The subgraphs (a–d) represent the reconstruction results of the phase-coordinate mapping model shown by the polynomial model, while the subgraphs (e–h) represent the reconstruction results of the proposed method. It is obvious that the former shows the situation of reconstruction mapping failure outside of the calibration zone, and the proposed method avoids this problem.

In addition, the reconstruction accuracy in the calibration and non-calibration areas are quantitatively analyzed in the form of planarity error and sphericity error by reconstructing planes and spheres at different positions. Figure 9 shows the reconstructed planar and spherical point cloud results and presents their error distribution in the form of a 2D error distribution map. In Table 2, we reconstruct the plane and sphere at three positions inside and outside of the calibration zone and calculate the fitted planarity error and sphericity error. Although there is a certain increase in the reconstruction error outside of the calibration area, it can still be used as an ideal result in full-field reconstruction scenarios without being completely unable to implement 3D reconstruction like the phase-coordinate mapping model.

To further explore the reconstruction ability of the proposed method outside of the calibration zone, we set a mask to continuously reduce its coverage range, thereby reducing the actual calibration zone. As shown in Figure 10a, the mask was set on the calibration board, and the area decreased continuously from mask1 to mask2 and then to mask3. We sampled the phase within the respective set range and calibrated the phase-angle model curves, as shown in Figure 10b–d. It can be seen from the results that although the calibration area was reduced, the model curve could still be extended in the unsampled area and assisted in inferring the 3D reconstruction outside of the calibration zone.

A 2D error distribution map of the reconstructed standard object outside of the calibration zone is shown in Figure 11a–f, and the subgraphs (a–c) show the reconstructed sphericity error as the mask setting changes. The subgraphs (d–f) show the reconstructed planarity error as the mask setting changes. To avoid randomness, we placed the standard object at three different positions outside of the calibration zone and calculated the average planarity error and average sphericity error separately. The experiment was conducted under each different mask setting, as shown in Figure 11g,h, and the results show the changes in the average planarity error and average sphericity error under different mask settings. It can be seen that the reconstruction accuracy outside of the calibration area is high, and that this calibration method does not cause significant fluctuations in error due to the reduction in the calibration area. This also proves that it can still achieve robust 3D reconstruction ability outside of the calibration zone.

## 5. Conclusions

In this paper, we propose a method to infer the 3D information of objects outside of the calibration zone using a phase-angle model for the scenario of unidirectional fringe projection. By sampling the phase within the local calibration area and obtaining the sample isophase planes through coordinate transformation and least squares estimation, it can be used for calibrating the phase-angle model and the rotation centerline. This method can achieve full-field 3D reconstruction, and its innovative points can be summarized as follows: 1. the 3D information outside of the calibration zone can still be obtained; 2. this method is efficient and concise, requiring only the projection of fringes at two calibration positions; 3. reconstruction can still maintain high accuracy and robustness on a global scale. The feasibility and efficiency of the proposed method were experimentally verified.

## Figures and Tables

**Figure 1 sensors-24-02581-f001:**
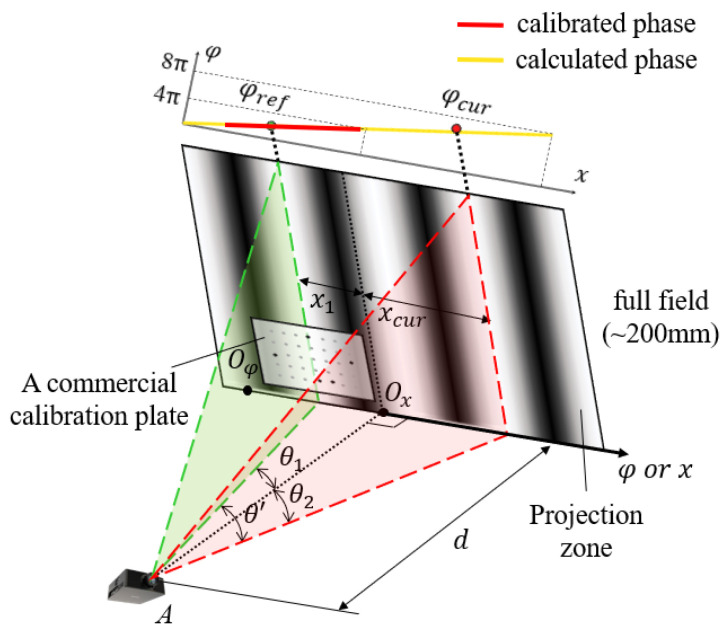
Schematic of the proposed global phase-angle relationship based on local calibration.

**Figure 2 sensors-24-02581-f002:**
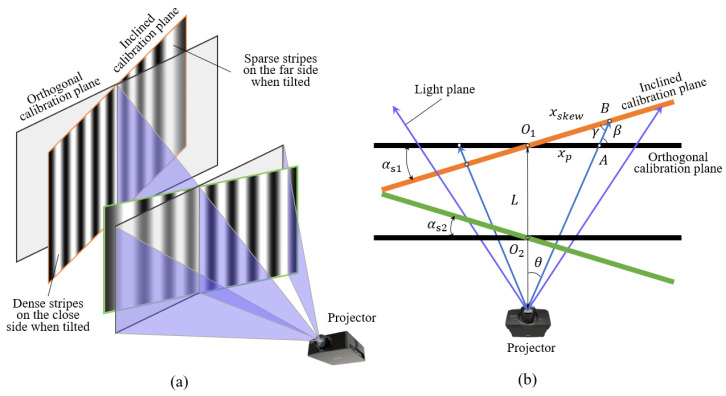
(**a**) Schematic of the geometrical position of the inclined calibration plane and the orthogonal calibration plane in the system; (**b**) top view of the positions.

**Figure 3 sensors-24-02581-f003:**
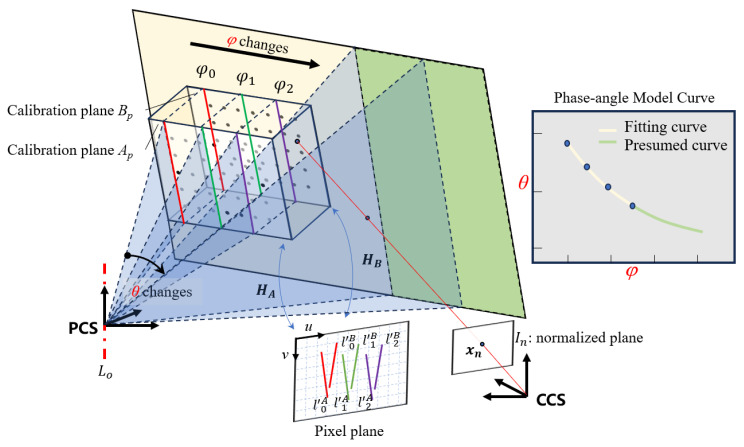
Schematic diagram of system calibration based on the phase-angle model.

**Figure 4 sensors-24-02581-f004:**
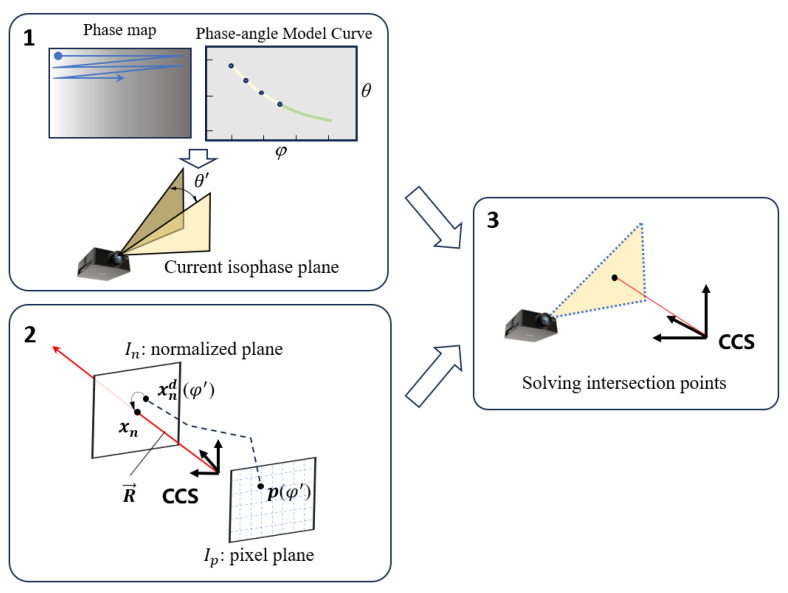
Reconstruction process diagram.

**Figure 5 sensors-24-02581-f005:**
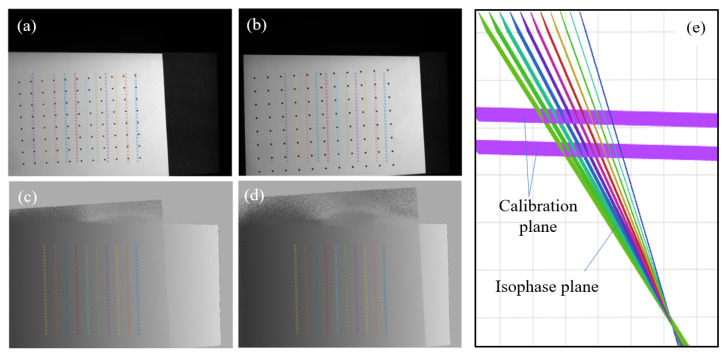
Calibration plate and phase sampling points in the phase maps. (**a**,**b**) are the distribution of sampling points on the original calibration images; (**c**,**d**) are the distribution of sampling points on the calibration plate phase maps; (**e**) is the distribution of calibration plane and isophase plane.

**Figure 6 sensors-24-02581-f006:**
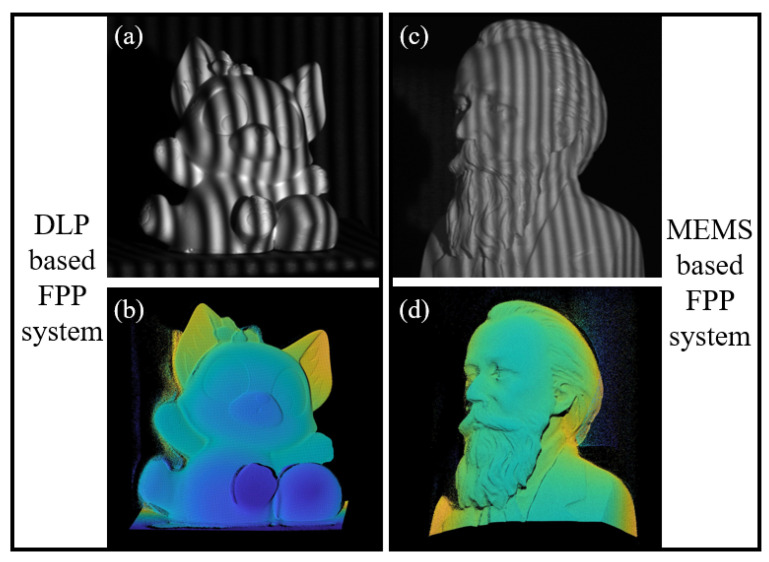
Results of applying the proposed method to reconstruct objects in different FPP systems. (**a**) is the object projected under DLP based FPP system; (**b**) is the reconstruction result under DLP based FPP system; (**c**) is the object projected under MEMS based FPP system; (**d**) is the reconstruction result under MEMS based FPP system.

**Figure 7 sensors-24-02581-f007:**
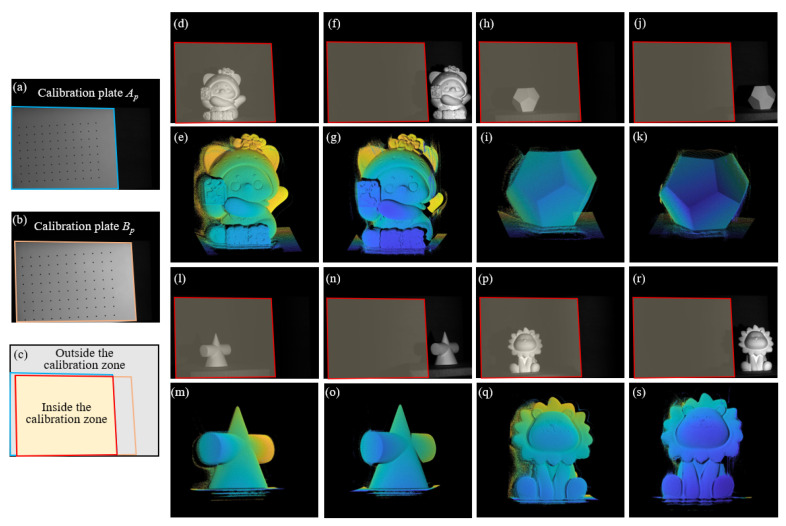
Reconstruction results of objects placed inside and outside of the calibration zone. (**a**,**b**) represent the positions of the two calibration boards; (**c**) is the overlapping area of the two calibration boards; (**d**,**h**,**l**,**p**) are the tested objects inside the calibration zone; (**f**,**j**,**n**,**r**) are the same objects outside the calibration zone; (**e**,**g**,**i**,**k**,**m**,**o**,**q**,**s**) are their corresponding reconstructed point cloud results.

**Figure 8 sensors-24-02581-f008:**
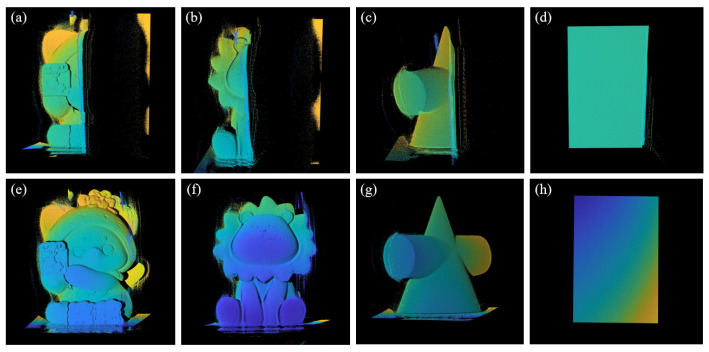
Comparison of reconstruction results between the phase coordinate mapping model and the proposed method. (**a**–**d**) represent the reconstruction results of the phase-coordinate mapping model shown by the polynomial model; (**e**–**h**) represent the reconstruction results of the proposed method.

**Figure 9 sensors-24-02581-f009:**
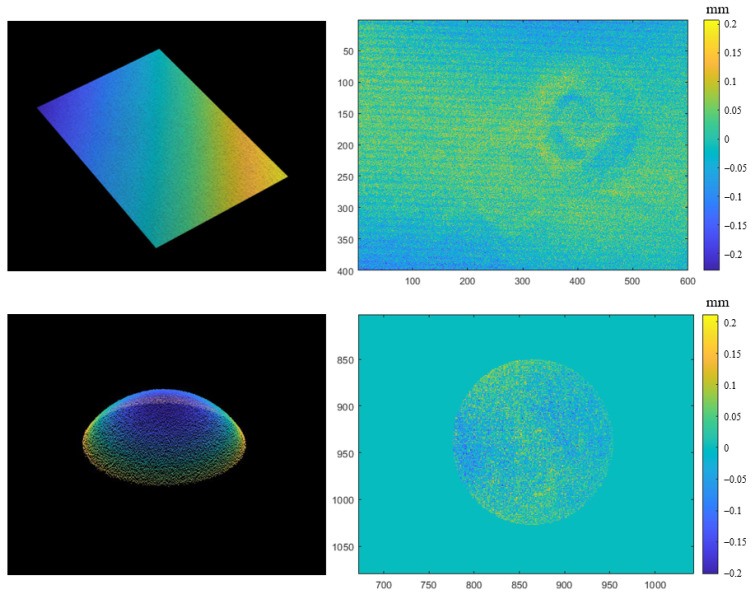
Point clouds and 2D error distribution maps for plane and sphere surfaces.

**Figure 10 sensors-24-02581-f010:**
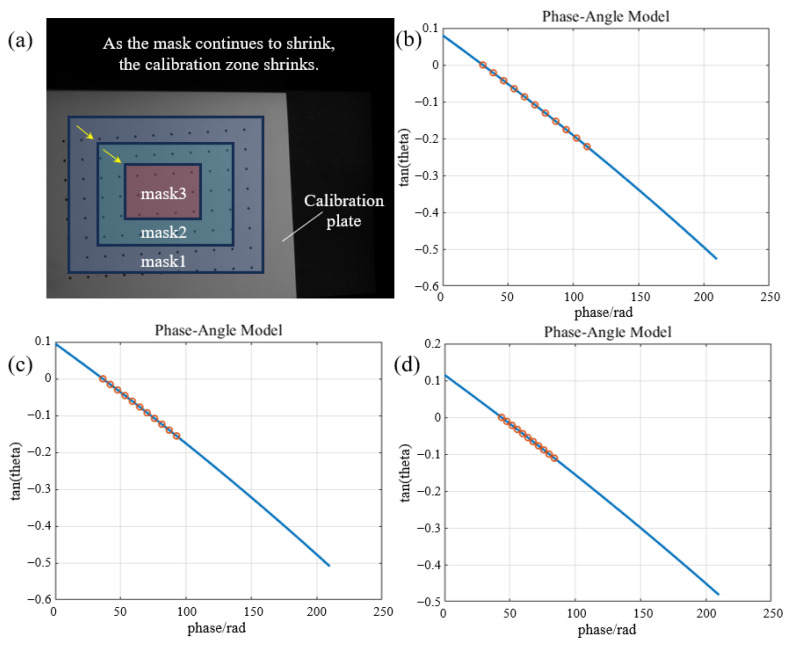
(**a**) Schematic of the masks for phase sampling on the calibration plate; (**b**–**d**) the phase-angle model curves under mask1, mask2, and mask3, respectively.

**Figure 11 sensors-24-02581-f011:**
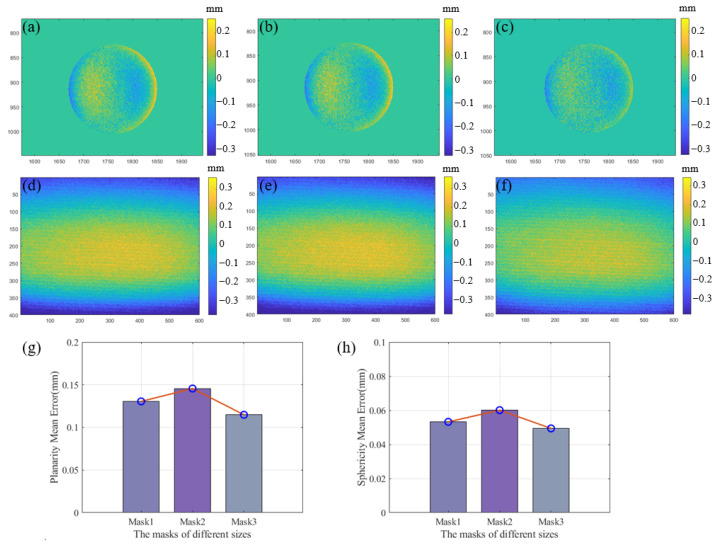
(**a**–**c**) Sphericity errors 2D distribution map outside of the calibration zone at mask1, mask2, and mask3 settings; (**d**–**f**) planarity error 2D distribution map at mask1, mask2, and mask3 settings outside of the calibration zone; (**g**) the planarity mean error variation at different mask settings; (**h**) the sphericity mean error variation at different masks settings.

**Table 1 sensors-24-02581-t001:** Phase-angle model-enabled calibration strategy parameters.

System Calibration Parameters		Value
Reference phase	$\varphi_{ref}$	30.6732
Reference isophase plane	$A^{F_{0}}$	0.9093
	$B^{F_{0}}$	−0.0134
	$C^{F_{0}}$	0.4160
	$D^{F_{0}}$	−139.7620
Phase-angle model	*a* _1_	0.2022
	*a* _2_	−382.4269
Rotational centerline	*v*_1_, *v*_2_, *v*_3_	(0.0268, 0.9993, −0.0259)
	*p*_1_, *p*_2_, *p*_3_	(97.6335, −184.3396, 116.7032)

**Table 2 sensors-24-02581-t002:** Planarity or sphericity errors in different zones.

Planarity or Sphericity Errors	Plane Inside Calibration Zone	Plane Outside Calibration Zone	Sphere, Inside Calibration Zone	Sphere, Outside Calibration Zone
Position1	0.0484 mm	0.1298 mm	0.0432 mm	0.0509 mm
Position2	0.0512 mm	0.1291 mm	0.0403 mm	0.0535 mm
Position2	0.0411 mm	0.1325 mm	0.0388 mm	0.0553 mm
Mean	0.0469 mm	0.1305 mm	0.0408 mm	0.0532 mm

## Data Availability

Dataset available on request from the authors.

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
