# Peer review of "Use of Phase-Angle Model for Full-Field 3D Reconstruction under Efficient Local Calibration"

_sensors, 2024, doi:10.3390/s24082581_

Round 1
Reviewer 1 Report
Comments and Suggestions for Authors
General comments
The article is devoted to improving the method of reconstructing 3D geometry using structured light. To reconstruct the geometry, the authors propose to use the phase angle model they developed, which allows one to obtain three-dimensional information about the geometry located outside the calibration zone. The proposed reconstruction method is based on the condition that the positions of a small (compared to the field of view) calibration plate ensure that two projection stripes overlap, which makes it possible to determine the relationship between the phase and the angle of deviation of the isophase plane, which is used to determine the location of the isophase plane in projection space, and not only in the area of the calibration plate. The experimental results presented in the article confirm the possibility of 3D reconstruction outside the calibration zone.
The article is of interest to specialists in the fringe projection profilometry. The main advantage of this article is the presentation of new experimental results using the 3D geometry reconstruction method developed by the authors.
The disadvantage of the article presented by the authors is the fact that the phase angle model and the corresponding reconstruction method have already been recently published by the authors (23. Lei, F.; Han, M.; Jiang, H.; Wang, X.; Li, X. A phase-angle inspired calibration strategy based on MEMS projector for 3D reconstruction with markedly reduced calibration images and parameters. Optics and Lasers in Engineering 2024, 176, doi:10.1016/j.optlaseng.2024.108078.). For this reason, originality/novelty is listed as “average”. Nevertheless, there is still novelty in the article, but it concerns only the results of experiments presented by the authors. It is the results of the experiments, described in detail by the authors, that are of practical importance for specialists involved in 3D reconstruction using the structured light.
Unfortunately, the quality of presentation leaves much to be desired. Let's start with the title. In my opinion, the title is not formulated quite correctly. Since the novelty of the article does not relate to the phase angle model as such, but to the method of full-scale 3D reconstruction using local calibration, the title should be formulated differently. I think the following option would be suitable: “Use of phase-angle model for full-field 3D reconstruction under efficient local calibration.” Or even shorter: “Full-field 3D reconstruction using efficient local calibration.” Also, there are a number of specific remarks.
Specific comments
Rows 84-87:
“There is currently a lack of a concise calibration strategy that can infer 3D reconstruction information beyond the range of the calibration plate for unidirectional fringe projections in reconstructing large-scale scenes.”
Why lack of? The strategy was already published by the authors in [23].
Row 126:
“Kc — are the camera's internal parameters”.
The camera's internal parameters are represented in equation (1) as a matrix with a number of variables which description is absent in the article. The description should be added.
Rows 129-130:
“… we can eliminate the lens aberration by the following equation:”
It is not quite correct formulation. The lens aberrations are not eliminated but only taken into account. So, it would be more correct to write as "we can take the lens aberration into account with the following equations".
Row 186:
“… and the two are at an angle of
”.
It is not clear what does “the two” mean here? May be incorrect English. Should be corrected.
Rows 323-326:
“Firstly, the correctness of the method was verified on the structured light system of the DLP projector; And use the same calibration strategy to test the applicability on emerging structured light projection devices, such as the FPP system of MEMS-mirror-based projectors.”
If there is firstly then where is secondly? Maybe it can be rewritten by the following way: " Firstly, the correctness of the method was verified on the structured light system of the DLP projector; And secondly, the same calibration strategy to test the applicability on emerging structured light projection devices, such as the FPP system of MEMS-mirror-based projectors was used." Or use some another formulation. Should be corrected.
Comments on the Quality of English Language
See specific comments.
Reviewer 2 Report
Comments and Suggestions for Authors
This manuscript proposed a local calibration model to realize a full-field reconstruction for fringe projection profilometry. The paper has great academic merit, and the proposed method is technically sound, but some revisions need to be brought to the manuscript to make it more impactful. My detailed comments are as follows:
(1) In Figure 11, it is recommended to use the same colorbar as (a) and (b) in (c), and to use blue markers of the same size as (h) in (g).
(2) In Figure 7, for the object with complex surface morphology in (d) and (f), it is evident that there are some stripe errors on the two ears and right face of the plaster model in the reconstruction results outside the calibration area. The quantitative analysis in Table 2 only compared simple objects such as planes and balls. It is recommended to add some experiments on complex shaped objects outside the calibration area.
(3) In Figure 11, it shows that the proposed calibration method does not cause significant fluctuations in error due to the reduction of the calibration area. I would like to know if there is an optimal calibration board size for this method.
Comments on the Quality of English LanguageNo obvious grammar errors,vocabulary richness needs to be improved
